# The Role of Transcription Factors in the Loss of Inter-Chromosomal Co-Expression for Breast Cancer Subtypes

**DOI:** 10.3390/ijms242417564

**Published:** 2023-12-16

**Authors:** Rodrigo Trujillo-Ortíz, Jesús Espinal-Enríquez, Enrique Hernández-Lemus

**Affiliations:** 1Computational Genomics Division, Instituto Nacional de Medicina Genómica, Mexico City 14610, Mexico; rodrigo.trujillo.vt@gmail.com; 2Center for Complexity Sciences, Universidad Nacional Autónoma de México, Mexico City 01010, Mexico

**Keywords:** breast cancer, gene co-expression networks, transcription factors, gene regulatory networks, breast cancer molecular subtypes

## Abstract

Breast cancer encompasses a diverse array of subtypes, each exhibiting distinct clinical characteristics and treatment responses. Unraveling the underlying regulatory mechanisms that govern gene expression patterns in these subtypes is essential for advancing our understanding of breast cancer biology. Gene co-expression networks (GCNs) help us identify groups of genes that work in coordination. Previous research has revealed a marked reduction in the interaction of genes located on different chromosomes within GCNs for breast cancer, as well as for lung, kidney, and hematopoietic cancers. However, the reasons behind why genes on the same chromosome often co-express remain unclear. In this study, we investigate the role of transcription factors in shaping gene co-expression networks within the four main breast cancer subtypes: Luminal A, Luminal B, HER2+, and Basal, along with normal breast tissue. We identify communities within each GCN and calculate the transcription factors that may regulate these communities, comparing the results across different phenotypes. Our findings indicate that, in general, regulatory behavior is to a large extent similar among breast cancer molecular subtypes and even in healthy networks. This suggests that transcription factor motif usage does not fully determine long-range co-expression patterns. Specific transcription factor motifs, such as CCGGAAG, appear frequently across all phenotypes, even involving multiple highly connected transcription factors. Additionally, certain transcription factors exhibit unique actions in specific subtypes but with limited influence. Our research demonstrates that the loss of inter-chromosomal co-expression is not solely attributable to transcription factor regulation. Although the exact mechanism responsible for this phenomenon remains elusive, this work contributes to a better understanding of gene expression regulatory programs in breast cancer.

## 1. Introduction

### 1.1. Breast Cancer Heterogeneity and Molecular Subtypes

Breast cancer is a complex disease characterized by significant heterogeneity, both at the histological and molecular levels. There is substantial variation in tumor characteristics, behavior, treatment response, and clinical outcomes observed among breast cancer patients.

Breast cancer molecular subtypes are a classification system that categorizes tumors based on their gene expression patterns or molecular profiles. The most widely used molecular subtyping system divides breast cancer into four major subtypes: luminal A, luminal B, HER2-enriched, and basal-like (also known as triple-negative breast cancer) [1,2]. Each subtype is associated with distinct molecular features, clinical behaviors, and responses to therapy [3].

These subtypes are linked to specific biological pathways, gene expression profiles, and genetic alterations. For instance, Luminal A tumors express hormone receptors typically have a more favorable prognosis, while basal-like tumors lack hormone receptors and tend to be more aggressive [4,5,6]. This diversity poses challenges for treatment decisions, emphasizing the need for personalized medicine [7].

High-throughput molecular profiling techniques reveal additional molecular subgroups within traditional classifications. For example, studies have identified distinct genetic alterations and outcomes within subtypes like the claudin-low subtype in basal-like breast cancer [8].

### 1.2. Gene Co-Expression Networks

Gene co-expression networks (GCNs) are computational models that map gene relationships based on their expression patterns across different conditions [9,10,11,12]. These networks quantify similarity or correlation in gene expression levels, portraying genes as nodes and edges to depict co-expression strength.

Various algorithms, including correlation-based methods and similarity measures, construct GCNs by assessing expression profile similarities and assigning weights to gene pairs, revealing systems-level gene interactions [13,14,15,16,17].

GCNs uncover genes involved in similar biological processes or shared regulatory mechanisms, aiding in understanding functional modules and regulatory circuits within biological systems [18,19,20].

Research focuses on identifying co-expressed gene modules within GCNs, grouping genes with correlated expression into functional clusters that often play roles in common biological pathways or processes [17,21,22]. These modules can be annotated with gene ontology terms or biological pathways to elucidate associated functions and pathways, shedding light on underlying biological mechanisms [23].

### 1.3. Loss of Long-Range Co-Expression in Cancer

Previous works from our group have used GCNs to analyze transcriptional regulation profiles in various carcinomas, noting two common phenomena in cancer GCNs: increased intra-chromosomal interactions and smaller network components compared to normal tissue GCNs. Termed loss of long-range co-expression, this phenomenon occurs consistently across cancer GCNs [9].

Despite efforts, the mechanism behind this phenomenon remains unclear. Copy number alterations’ involvement was explored but was found to have an non-significant impact on co-expression [23,24]. Investigation into microRNAs revealed minimal influence, observed only in specific regions like the DLK-DIO3 region in breast cancer [25,26]. For clear cell renal carcinoma, in turn, we did not observe miRNA expression biasing the co-expression to neighboring genes [27].

Finally, we also investigated whether the methylation profile of DNA may be involved in this loss of long-range co-expression in cancer. We showed that, despite the fact that methylation profile does exert influence on particular genes that may affect the cell states depending on the progression stage [28], we cannot establish a clear association between methylation profiles and co-expression between neighboring genes.

### 1.4. Transcription Factors Influence in GCNs

Given that this seems to be a local effect, we decided to investigate whether transcription factors and their binding motifs may exert influence on the transcription of close genes with a similar expression pattern. To this end, we investigated the role of transcription factor binding sites in the gene co-expression networks for breast cancer subtypes. We developed a computational pipeline that infers gene co-expression networks from a gene expression matrix. Then, we split the networks into communities according to the Louvain algorithm. Detected communities were enriched to the transcription factors that may regulate the genes from the same community.

Once transcription factors were inferred, we built a TF-target gene regulatory network for each breast cancer subtype, as well as for the control group. We compared the number of TF/motifs observed on each network, and we calculated their topological parameters. Finally, we analyzed the main transcription factor motifs, as well as those unique for each GRN.

## 2. Results

### 2.1. Gene Co-Expression Networks

In Figure 1 we show the top 3500 gene co-expression interactions in healthy (A) and Basal (B) cases. As observed, gene co-expression networks in cancer show a clear loss of long-distance co-expression. The large majority of genes interact with other genes but from the same chromosome, as described in [9].

Figure 1C,D shows the same networks, but the genes are painted according to the Louvain community to which each gene belongs. To note, in the case of healthy network painted by community, genes are clearly grouped; meanwhile, for the case of Figure 1A, genes interact with other genes from different chromosomes. Therefore, in the control network, communities are not biased by the chromosome location of their genes.

### 2.2. TF-Target Network Topology Is Similar between Cancer Subtypes

Next, we decided to investigate whether TFs may influence the global co-expression patterns for each phenotype. That is the reason for which we constructed a TF-target gene regulatory network (GRN) for the breast cancer subtypes, as well as for control phenotype (Figure 2). In this figure, all breast cancer GRNs are composed of one giant component, which includes all the enriched TFs and targets of them. A resume of the topological features for each GRN is shown in Table 1.

As can be noticed, the GRN for control has the smallest number of TFs and nodes. On the other hand, the network with more TFs and genes correspond to the Luminal A subtype. The TF-targets GRNs were inferred using the communities detected in the gene co-expression networks, represented in Figure 1. Our main hypothesis was that the single-chromosome components observed in Figure 1B may be predominantly regulated by a small set of specific TFs. Additionally, each component would be regulated by specific TFs as well. As observed, this hypothesis has been indeed rejected: communities detected by the Louvain algorithm are not biased to specific TFs; instead, the large majority of genes are regulated by an important amount of quite general TFs.

There are of course, a few exceptions, as it can be appreciated in the small star-like structures in all breast cancer subtype GRNs. To note, all those structures in the breast cancer GRNs are composed of genes from one single chromosome. On the other hand, the same star-like structures also appear in healthy GRNs, but in that case, genes are located in different chromosomes.

### 2.3. TFs Are Strongly Shared between Cancer Subtypes

After analyzing the topology of the five GRNs, we compared the number of gene targets for each network. The results are depicted in Figure 3A, where the distribution of targets for each GRNs are shown. A remarkable difference between healthy GRN targets compared with any breast cancer subtype becomes evident in the upper part of the distribution, i.e., the TFs with more targets are part of the cancer networks. For instance, the TF with more targets for healthy GRNs is *ZF5*, with 214 targets. In the case of Luminal A, *ZF5* regulates 714 targets, 3.5 times more than in the control case.

Taking into account that all cancer GRNs present a similar number of target genes, we intersected all sets of TFs to observe whether or not those TFs are unique or shared. The results are shown in Figure 3B. The Venn diagram presented here shows that only a few TFs are unique for each GRN, and more importantly, the large majority of TFs are strongly shared, in particular for cancer phenotypes. The red squares highlight the number of TFs shared by all phenotypes (37), as well as those shared by cancer-only GRNs (86). This last result reflects two main facts: (i) the TF-target structure is very similar in breast cancer, independent of the subtype, and (ii) in the non-cancer GRN, despite sharing the majority of its TFs, the number of TFs associated with the co-expression network is much smaller than in cancer.

Despite this, the number of shared targets is very low compared with the amount of unique targets for each phenotype, as seen in the right part of Figure 3B. There is just one target shared by all TF-target networks (GPAA1), only 63 targets shared by the four breast cancer subtypes.

### 2.4. Gene Targets for the Cancer Subtypes Share Regulation Patterns with the Non-Cancer GRN

Once the topological features for each GRN were calculated, and the number of common and unique TFs per network were obtained, we analyzed the TFs with more regulated genes. In Table 2 the top-10 TFs for each GRN are placed. Additionally, in the table are also shown the motif for each TF, as well as their number of regulated targets.

As noticed, transcription factor *ZF5* appears in the top 10 regulators in all cases, but in Luminal B GRN, (*ZF5* appears in the 18th place, with 266 targets). However, the number of target genes is larger in any breast cancer GRN compared with control. An additional feature is observed in the motifs appearing in Table 2; there is a common heptameric sequence in several TFs: CCGGAAG. In fact, this sequence is the most common one in the whole TF-target universe for the five GRNs. The sequence CCGGAAG appears in TFs that regulate 5936 targets in Basal GRN, since the sequence appears in 28 of the TFs. For instance, the transcription factor *Elk-1* has three different motifs in Table 2: RCCGGAAGTGN for control, RACCGGAAGTR for Luminal A, NNCCGGAAGTN for Luminal B, and NRSCGGAAGNN for HER2 and Basal (notice that this one does not contain the aforementioned heptamer).

Furthermore, the majority of these top 10 TFs are the same for any phenotype, which is in agreement with the Venn diagram of Figure 3B. This fact reflects that the transcriptional regulation in normal cells and breast cancer cells is very similar in terms of the nature of the transcription factors, as well as their number of regulated genes. The latter is also corroborated by the similarity observed in the mean values of the boxplot depicted in Figure 3A. Even if the distribution is narrower in control, all GRNs have similar mean values.

### 2.5. Transcriptional Targets in Cancer Have More Regulators than in Control

Despite the similarity in terms of number of targets and the similarity between TFs in cancer and control GRNs, the regulated genes in cancer have much more TFs influencing their expression. Figure 4 shows the number of TFs that regulate each target gene for basal and control GCNs.

In Figure 4, TFs are depicted in the upper part (red diamonds), while targets are the circles colored by the chromosome they belong to. Additionally, genes are placed along the *X* axis according to the number of regulatory TFs for each gene, i.e., the more TFs connected to a gene, the more placed to the right. For that reason, some genes might be overlapped if they belong to the same chromosome and also have the same number of TF regulators. Other relevant features observed in the figure is that targets in cancer do not belong to all chromosomes. On the contrary, for control GRN, genes from all chromosomes are regulated by the TFs.

### 2.6. Unique TFs for Each Subtype Regulate Chromosome-Specific Target Genes in Cancer

Finally, we analyzed the unique TFs observed in the Venn diagram of Figure 3B, since they are very few for all phenotypes under study, and they may provide hints on the transcriptional landscape for each subtype.

Figure 5 shows the unique TFs for each GRN, as well as their targets. From this figure, we can observe several features:All unique TFs have a small number of regulated targets, which is again in agreement with the high similarity in the top TFs (Table 2 and Figure 3B).All unique TFs in cancer regulate genes from the same chromosome, except *NF-Y* and *YB-1* for Luminal A GRN.In the control GRN, the TFs regulate genes from any chromosome.

For Basal subtype, unique TFs only regulate genes from Chr. 1, 17 and 19, with Chr 17 being the one with the most regulation (all but two genes in that chromosome, regulated by *ZNF445*).

In contrast, for the Her2+ GRN, the 32 unique TFs regulate one third of the total amount of genes. Interestingly, those TF-target interactions become star-like structures (Figure 5).

For Luminal B, *ATF4*, a unique TF regulating this subtype also regulates genes that are only regulated by that gene, thus indicating a specific action in that phenotype.

## 3. Discussion

In this work, we have constructed transcription factor (TF)-target gene regulatory networks (GRNs) for the four breast cancer subtypes, namely, Luminal A, Luminal B, Her2+ and Basal-like, as well as for non-cancer phenotypes. To construct these regulatory networks, we have used gene co-expression networks (GCNs) grouped by communities, which in turn were calculated with the Louvain method. We inferred the TFs that could regulate each community according to the Transcription Factor Binding Site (TFBS) that matched with the genes of that community.

All the methodologies implemented and explained in this work were performed with the objective of investigating whether or not TF usage is significantly involved in the phenomenon of loss of long-range co-expression [9]. Since the phenomenon has been observed clearly in breast cancer and breast cancer subtypes, research involving these subtype’s results are appealing. Additionally, so far, there is no mechanistic explanation of this well-established phenomenon. In other words, we wanted to investigate whether TF usage influences the loss of long-range co-expression or the gain of short-distance interactions. If that were the case, specific TFs would regulate the target genes, i.e., a TF would preferably regulate a (local) cluster of genes.

In the case of our TF-gene regulatory networks (Figure 2), we can observe that TFs in cancer do not preferentially regulate genes from a certain chromosome, except for the specific case of Chr10 in Luminal B and Basal subtypes, as well as Chr 13 in Luminal A. Genes from chromosome 17 also appear regulated by specific TFs. However, the rest of the targets are regulated by a large set of TFs. The latter result allows to us to suggest that transcription factor regulatory dynamics does not influence the loss of long-range co-expression in cancer. Therefore, something else would do it.

From Figure 3B, we can observe that the large majority of TFs are shared between phenotypes. On the one hand, the TFs shared between all cancer subtypes and control network (37) correspond to almost 80% of all TFs in the control GRN. On the other hand, the 86 shared TFs between the cancer subtypes also reflects a very common transcriptional regulation in cancer.

Additionally, the important difference between TFs regulating cancer and non-cancer networks also suggest that in cancer, several processes have been activated. Those processes do not appear in normal phenotypes, and in addition, they are common for (at least) breast cancer.

This is also in agreement with a previous study developed by our group [23]. There, a community detection analysis was performed in the Luminal A breast cancer subtype GCN. Communities found in the Luminal A network enriched with a high significance ($p_{val}<1\times 10^{-10}$) to processes associated with the cell cycle. In particular, a community was composed of genes from different chromosomes where the *FOXM1* transcription factor regulated all genes from that community. Other TFs appearing in that community were *NUSAP, CENPA, HJURP*, and *RAD51* (Figure 6A). In addition, all genes from that community were highly overexpressed with respect to the control.

Interestingly, in this case, the aforementioned genes were found to be regulated by 7 TFs: *E2F-2, E2F-3:HES-7, E2F-4, YB-1, NF-Y, NF-YA* and *ZF5*. Perhaps those TFs are acting upstream of the aforementioned ones; after that, *FOXM1, NUSAP, CENPA*, and the other TFs may provoke the overexpression of their targets. Taking into account that the community observed in the Luminal A GCN is composed of genes from different chromosomes, the results presented here indicate that TFs better explain the inter-chromosome co-expression much clearer than intra-chromosome interactions.

Finally, each regulatory network posses a small group of unique TFs. In all cancers, those unique TFs per phenotype regulate only a reduced number of genes from the same chromosome (Figure 5). Conversely, in controls, those unique TFs do have targets from any chromosome. This also could indicate that for cancer, unique TFs may exert a reduced but specific influence. This last may explain only partially the strongly connected intra-chromosome communities found in breast cancer subtype GCNs.

The reduced number of chromosomes being regulated by TFs in the Basal GRN observed in Figure 4, which contrasts importantly with the GRN observed for control, indicates that the TFs in the control, despite being almost the same than those for the cancer GCNs, exert their influence more distributively than their cancerous counterparts. Perhaps the number of TFs necessary to be regulated is better optimized than in cancer. On the contrary, for the Basal GCN, one can argue that genes need to be more regulated in order to accomplish their function.

Another noteworthy finding is that genes from chromosomes 8 and 17 are strongly regulated, much more so than genes from other chromosomes. This could be related to the fact that genes in chromosomes 8 and 17 have been widely reported to be dysregulated in cancer. In fact, Chrom. 17 amplicon is one of the best-known DNA alterations in breast cancer [29,30]; for example, one of the subtypes (HER2+) receives its name from the alterations in a gene placed in Chr17q12.

The last result is also in agreement with previous work from our group. In [23], we demonstrated the ubiquity of a barely connected cluster of genes in breast cancer subtype GCNs—in particular, a cluster from Chr8q24.3 appeared in the four subtypes. The latter did not happened for the control GCN. However, in this case, the TFs strongly regulating genes from chromosomes 8 or 17 also regulate genes from other chromosomes. Therefore, the appearance of strongly connected communities from one chromosome, and more importantly, from the same cytoband, should be triggered or promoted by other mechanism.

Finally, regarding the CCGGAAG sequence observed to be repeated in the top-TF motifs for all GRNs reinforces the fact that transcription factor regulation is an evolutionary mechanism that minimizes the size of regulatory sequences. CCGGAAG is not the only repeated sequence. For instance, GGGCGGG or GCGCGCG are also frequently observed.

## 4. Materials and Methods

### 4.1. Gene Expression Data

TCGA, The Cancer Genome Atlas, is a vast and accessible database offering comprehensive genomic, transcriptomic, and clinical data for cancer research, including breast cancer. It encompasses diverse clinical information like patient demographics, tumor details, treatments, and survival outcomes [3,31].

Through high-throughput sequencing, TCGA enables the exploration of gene expression patterns, genomic alterations, and variations in breast cancer. This data aids in deciphering the complex molecular landscape, identifying potential therapeutic targets, and investigating molecular/clinical associations.

RNA-Seq data from The Cancer Genome Atlas was downloaded in HTSeq-Counts format from the Genomic Data Commons (GDC) Data Portal for the four different breast cancer molecular subtypes (according to the PAM50 classifier): Luminal A, Luminal B, Her2+ and Basal. Samples tagged as *Primary Tumor* constitute the cancer data set, while the *Normal* data set is integrated by samples with sample type: Solid Tissue Normal, which is healthy tissue adjacent to the tumors from some of the same individuals considered as cases.

Gene expression data pre-processing and differential expression was carried out by following a pipeline adapted from [9]. The SnakeMake workflow for such analysis can be found at https://github.com/ddiannae/tcga-xena-pipeline (accessed on 2 December 2023).

TCGA samples were annotated using their annotation reference file, GENCODE v22, https://gdc.cancer.gov/about-data/gdc-data-processing/gdc-reference-files (accessed on 2 December 2023). Only protein coding genes, conventional chromosomes and genes that appear in the GENCODE v37 (April, 2021) annotation, were kept. After that raw count matrices were assembled and integrated with their corresponding annotations.

The NOISeq [32,33] R library (https://www.bioconductor.org/packages/release/bioc/html/NOISeq.html, accessed on 2 December 2023) was used for quality control. Genes with low expression values were removed by identifying genes with a count mean expression < 10 and genes with a zero-count expression value in more than 50% of the samples. Samples were removed from further analysis if their mean expression value fell two standard deviations over or under the total mean.

Bias detection plots were obtained to assess mean gene length and %GC and RNA content bias, and a combination of different within/between normalization strategies were tested to remove bias presence using the EDASeq [34] package (https://bioconductor.org/packages/release/bioc/html/EDASeq.html, accessed on 2 December 2023). The best alternative for each tissue was selected after visual inspection of the NOISeq plots. Principal Component Analysis plots showed overlap between experimental groups; therefore, ARSyNSeq, also from the NOISeq library, was used for multidimensional noise reduction and batch effect removal with default parameters.

### 4.2. Co-Expression Network Inference and Network Modularity Calculations

Network inference was carried out by calculating the pairwise mutual information measure for all genes using a custom-made parallel implementation of the ARACNe algorithm [35,36,37]. Mutual information thresholds were set to include only the top 100,000 larger mutual information co-expression interactions. A *Singularity* (https://sylabs.io/, accessed on 2 December 2023) container was created to run ARACNE inside the Snakemake workflow. The source code can be found here: https://github.com/ddiannae/ARACNE-multicore (accessed on 2 December 2023).

Network modularity calculations were performed by using the Louvain algorithm [38]. We conserved the top 20 communities according to the size of those communities. SnakeMake code for network inference and modularity calculations can be found at https://github.com/ddiannae/distance-analysis (accessed on 2 December 2023).

### 4.3. Transcription Factor Binding Motif Analysis and TF-Target Network Construction

Transcription factor binding motif analysis was made using g:profiler [39] using both the web service (https://biit.cs.ut.ee/gprofiler/gost, accessed on 2 December 2023) and the gprofiler2 R package (https://cran.r-project.org/web/packages/gprofiler2/vignettes/gprofiler2.html, accessed on 2 December 2023). For all hypergeometric tests, we considered significant those hits with FDR corrected *p*-value <0.05. Transcription binding motif analysis was carried out by looking at regulatory motif matches from the *TRANSFAC* [40] database (http://genexplain.com/transfac/, accessed on 2 December 2023), which also includes a comprehensive list of human transcription factors.

Once the TFBS analysis was performed, a TF-target network was created for each phenotype by linking the TF with their respective targets.

## 5. Conclusions

Transcriptional regulation is one of the fundamental mechanisms to sustain life. It is a general process for all known species, and it is indeed one of the most carefully preserved phenomena, both in prokaryots and eukaryots, respectively. In this work, we have observed subtle differences in the TF-target regulation between breast cancer subtypes and control GRNs. Despite those differences, the broader form of TF-target behavior remains essentially the same.

With the latter in mind, we can argue that the phenomenon of loss of long-range co-expression observed in several types of cancer is not predominantly due to a biased form of transcriptional regulation, at least not in the context of TF motif usage. A few examples of unique TF-target observed in cancer subtypes do not explain the dramatic difference in the intra-chromosome co-expression interactions between cancer and non-cancer networks.

Further research to understand the mechanisms behind the loss of long-distance co-expression is necessary. However, it is important to mention that this work serves to determine that the phenomenon is not predominantly related to the transcription factor motif usage and binding target regulation.

## Figures and Tables

**Figure 1 ijms-24-17564-f001:**
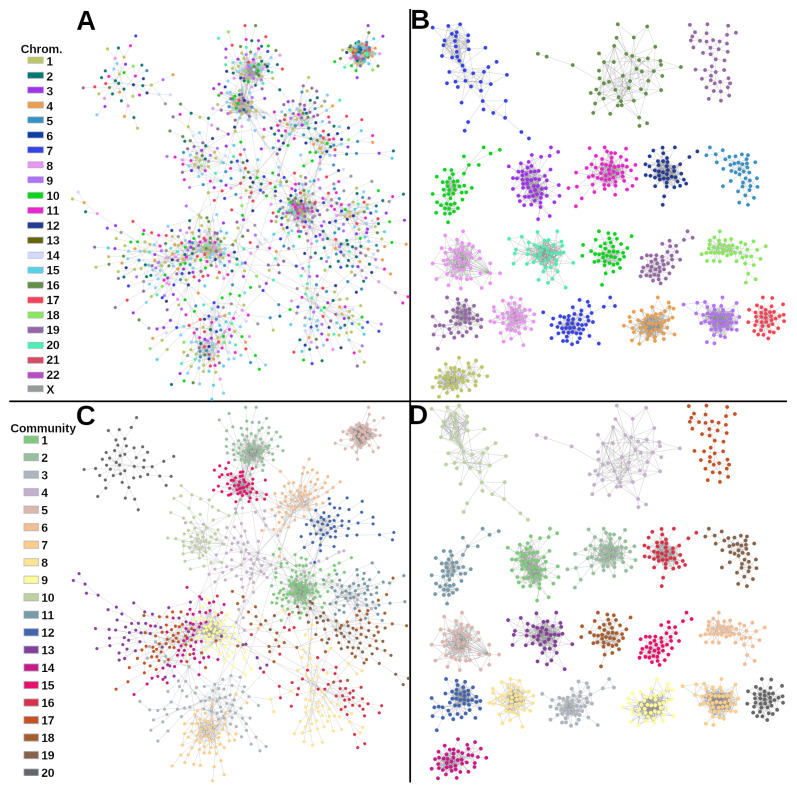
(**A**) Co-expression networks for control and Basal phenotypes. (**A**,**B**) In these networks, genes are colored by the chromosome in which they are located. (**C**,**D**) Genes are colored by the community in which they appear. GCN for Basal phenotype (**B**,**D**) shows a clear aggregation for chromosome and communities, while for control GCN, the network in (**A**) is not grouped by chromosome.

**Figure 2 ijms-24-17564-f002:**
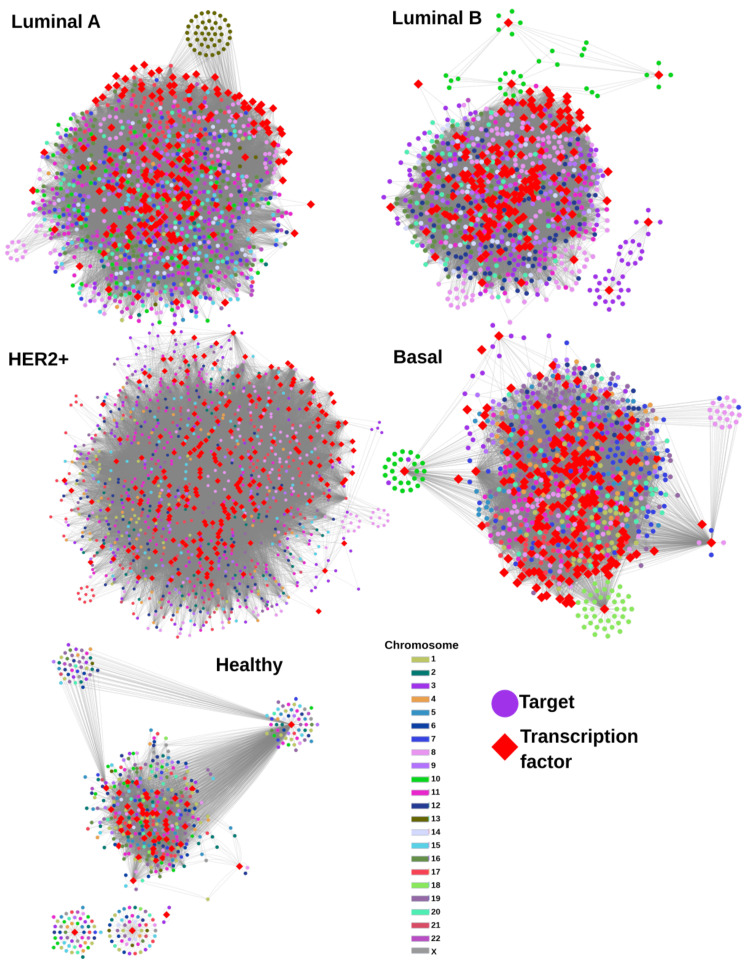
TF-target Gene Regulatory Networks (GRNs) for all phenotypes. TFs are represented by red diamonds, and targets are circles colored by the chromosome where they are located.

**Figure 3 ijms-24-17564-f003:**
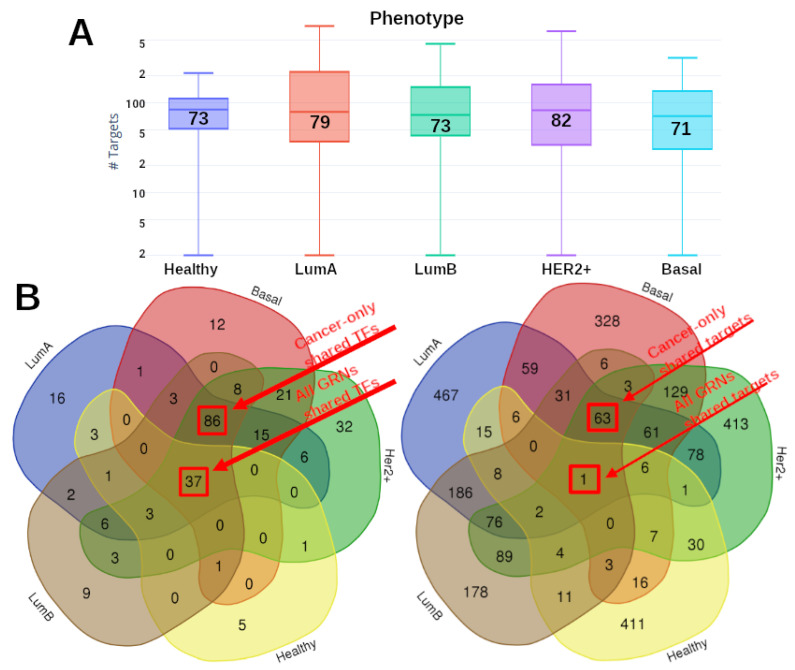
(**A**) Distribution of number of targets for each GRN. Numbers inside the box represent the mean values. The *Y* axis shows, in a log scale, the number of targets regulated by the TFs for each phenotype. To note, the distribution of cancer networks is wider than the control one. (**B**) Venn diagrams of the shared TFs and the shared targets by the five phenotypes.

**Figure 4 ijms-24-17564-f004:**
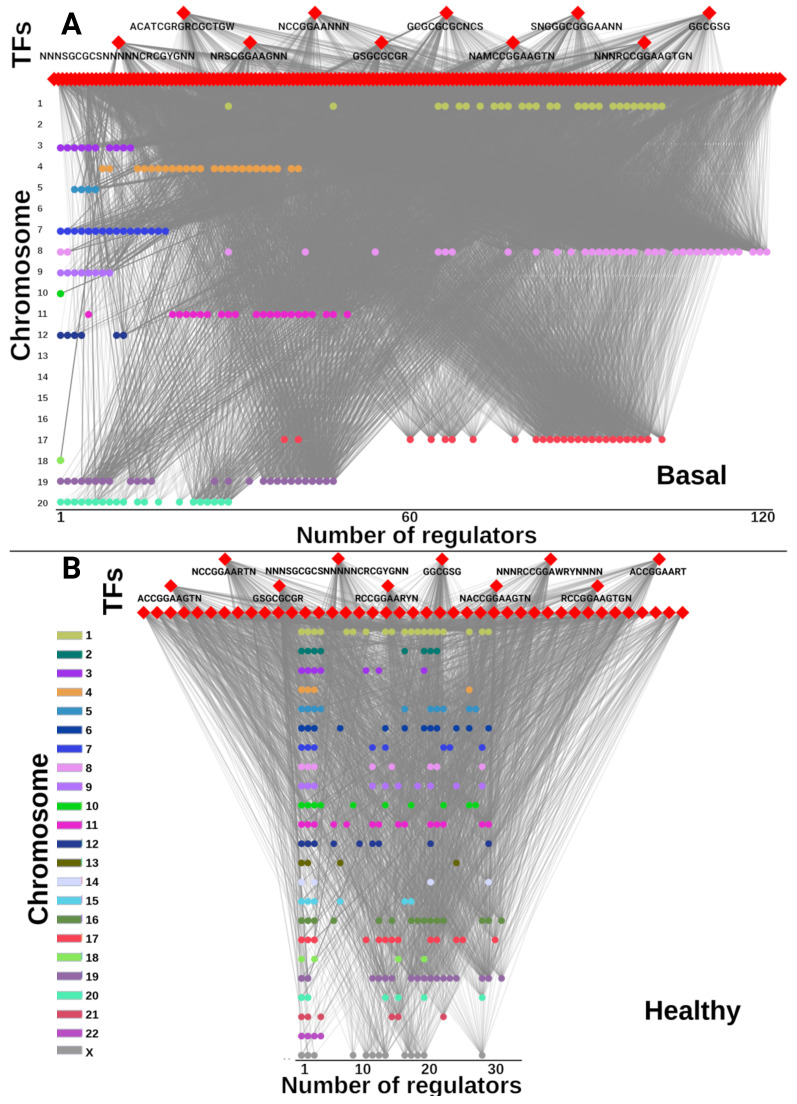
Number of regulated targets by TFs. (**A**) Basal GRN. (**B**) Control GRN. Target genes are colored by chromosome, and the *X*-axis location of target genes is proportional to the number of regulators.

**Figure 5 ijms-24-17564-f005:**
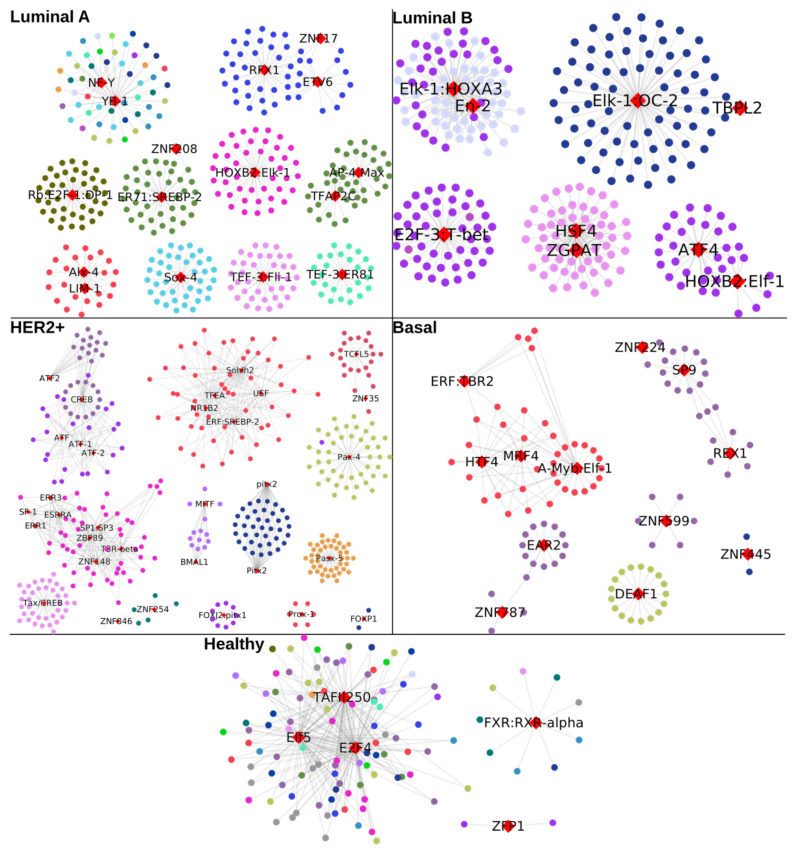
Unique TFs for each GRN. Each network shows the targets regulated by those phenotype-exclusive TFs. Notice that for any breast cancer subtype, the majority of targets for any TF are located at the same chromosome. Color code is the same than Figure 4.

**Figure 6 ijms-24-17564-f006:**
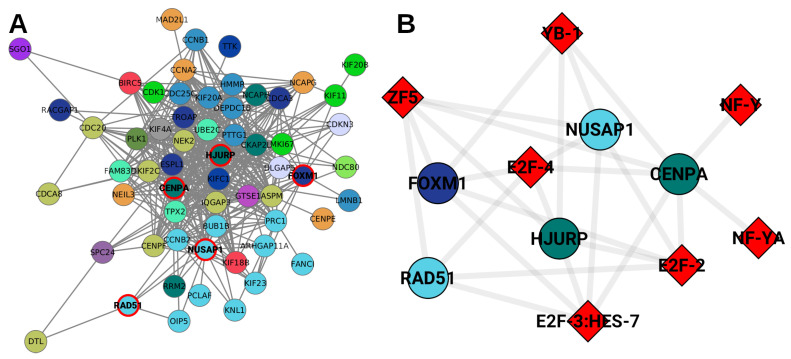
Luminal A networks. (**A**) A network community extracted from the gene co-expression network. Genes from this network are associated with cell cycle. TFs are circled in red. (**B**) Red diamonds correspond to TFs regulating the red-circled TFs from (**A**). Genes are colored according to the chromosome where they are located.

**Table 1 ijms-24-17564-t001:** Topological features of the five TF-target GRNs obtained from the top 20 communities of the five gene co-expression networks.

Phenotype	Nodes	Links	Targets	TFs
Healthy	572	6962	521	51
Luminal A	1239	51,439	1060	179
Luminal B	820	33,343	661	159
HER2+	1181	46,965	963	218
Basal	903	29,118	719	184

**Table 2 ijms-24-17564-t002:** Topological features of the five TF-target GRNs obtained from the top 20 communities of the five gene co-expression networks. Red sequences correspond to the same heptamer, CCGGAAG.

Phenotype	TF	# Targets	Motif
	ZF5	214	GSGCGCGR
	E2F	188	GGCGSG
	ER81	158	RCCGGAARYN
	Elk-1	155	R**CCGGAAG**TGN
Control	E2F-3:HES-7	152	NNNSGCGCSNNNNNCRCGYGNN
	ELF4	145	NCCGGAARTN
	Erg	142	A**CCGGAAG**TN
	PEA3	134	NA**CCGGAAG**TN
	c-Ets-1	132	NNNRCCGGAWRYNNNN
	ERG	129	ACCGGAART
	ZF5	714	GGSGCGCGS
	ER81	688	RCCGGAARYN
	Elk-1	651	RA**CCGGAAG**TR
	Erg	650	NACCGGAARTN
Luminal A	E2F-4	626	NTTTCSCGCC
	PEA3	599	NA**CCGGAAG**TN
	Elf-1	593	NANGCGGAAGTN
	Fli-1	578	NACCGGAARTN
	ERG	530	ACCGGAARTN
	E2F-1	502	NGGGCGGGARV
	Elk-1	453	NN**CCGGAAG**TN
	Elf-1	405	NNAN**CCGGAAG**TGS
	ER81	405	NN**CCGGAAG**YG
	PEA3	397	R**CCGGAAG**YN
Luminal B	Erg	347	NACCGGAARTN
	ELK-1	346	A**CCGGAAG**TN
	Fli-1	344	NACCGGAARTN
	ERG	339	ACCGGAARTN
	GABP-alpha	338	NR**CCGGAAG**TN
	Erm	333	NNSCGGAWGYN
	ZF5	627	GSGCGCGR
	E2F-4	566	SNGGGCGGGAANN
	E2F	535	GGCGSG
	E2F-3:HES-7	472	NNNSGCGCSNNNNNCRCGYGNN
HER2+	E2F-2	464	GCGCGCGCNCS
	E2F-1	423	NKTSSCGC
	pax-6	421	NYACGCNYSANYGMNCN
	Sp1	409	GGGGCGGGGC
	Elk-1	391	NRSCGGAAGNN
	GABP-alpha	378	NNNR**CCGGAAG**TGN
	ZF5	316	GSGCGCGR
	E2F-2	311	GCGCGCGCNCS
	Elk-1	304	NRSCGGAAGNN
	E2F	299	GGCGSG
Basal	GABP-alpha	285	NNNR**CCGGAAG**TGN
	E2F-3:HES-7	270	NNNSGCGCSNNNNNCRCGYGNN
	Elf-1	265	NAM**CCGGAAG**TN
	TCF-1	258	ACATCGRGRCGCTGW
	E2F-4	247	SNGGGCGGGAANN
	ETV7	238	NCCGGAANNN

## Data Availability

Data is contained within the article.

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
