# Peer review of "The Role of Transcription Factors in the Loss of Inter-Chromosomal Co-Expression for Breast Cancer Subtypes"

_ijms, 2023, doi:10.3390/ijms242417564_

Round 1

Reviewer 1 Report

Comments and Suggestions for Authors

Summary: Goal of the manuscript is to reveal the underlying differences of regulatory mechanisms that govern gene expression patterns among the 4 major subtypes of breast cancers (Luminal A, Luminal B, HER2+, and Basal) versus normal breast. Thus, they identified gene co-expression networks (GCNs) across different samples whereas a clear loss of long-distance co-expression (inter-chromosome co-expression interactions) was shown across all the breast cancer samples. Most genes interact with other genes are from the same chromosome in cancers. They calculated TFs that regulate the communities in each GCN and found a large similarities of regulation behavior among all the samples, indicating that most of genes were regulated by a common TF set. TF binding motifs are similar as well. Thus, the TF binding motifs do not explain the loss of long-range co-expression patterns in cancers. Next, they examined the co-expression networks influenced by the common usage TFs for each group. Notably, despite the fact that the normal group shares the majority of its TFs, the number of TFs associated with the co-expression network is much smaller whereas TF-target network is way bigger in cancers and the topology is similar between each subtype. On the other hand, the regulated genes in cancers have more TFs that can influence their expression than in control. Next, they compare the TF’s targets in subtypes and the healthy. They found only a few TFs are unique for each GRN; most TFs are strongly shared, for cancer phenotypes. Finally, they analyzed the unique TFs in each subtype and concluded that all unique TFs have a small number of regulated targets and regulate genes from the same chromosome (with few exceptions). In control GRN, TFs regulates genes from any chromosome.

Overall, this manuscript is well-written and innovative. BC subtypes have complicated underlying regulation. Study the global networks is essential. The manuscript discussed about the limitation of the study and the meaning of the results. The result is interesting. Here are some questions:

1.     Most genes interact with other genes are from the same chromosome in cancers. Is this a cancer-universal phenomenon or breast cancer specific?

2.     Why genes on the same chromosome often co-express and why loss of long-range co-expression patterns in cancers remain unclear?

3.     Are the genes controlled by health TFs also controlled by the same set of TFs in BC? What kind of pathways are they involved in?

Author Response

Summary: Goal of the manuscript is to reveal the underlying differences of regulatory mechanisms that govern gene expression patterns among the 4 major subtypes of breast cancers (Luminal A, Luminal B, HER2+, and Basal) versus normal breast. Thus, they identified gene co-expression networks (GCNs) across different samples whereas a clear loss of long-distance co-expression (inter-chromosome co-expression interactions) was shown across all the breast cancer samples. Most genes interact with other genes are from the same chromosome in cancers. They calculated TFs that regulate the communities in each GCN and found a large similarities of regulation behavior among all the samples, indicating that most of genes were regulated by a common TF set. TF binding motifs are similar as well. Thus, the TF binding motifs do not explain the loss of long-range co-expression patterns in cancers. Next, they examined the co-expression networks influenced by the common usage TFs for each group. Notably, despite the fact that the normal group shares the majority of its TFs, the number of TFs associated with the co-expression network is much smaller whereas TF-target network is way bigger in cancers and the topology is similar between each subtype. On the other hand, the regulated genes in cancers have more TFs that can influence their expression than in control. Next, they compare the TF’s targets in subtypes and the healthy. They found only a few TFs are unique for each GRN; most TFs are strongly shared, for cancer phenotypes. Finally, they analyzed the unique TFs in each subtype and concluded that all unique TFs have a small number of regulated targets and regulate genes from the same chromosome (with few exceptions). In control GRN, TFs regulates genes from any chromosome.

Overall, this manuscript is well-written and innovative. BC subtypes have complicated underlying regulation. Study the global networks is essential. The manuscript discussed about the limitation of the study and the meaning of the results. The result is interesting. Here are some questions:

 The authors want to thank Reviewer 1 for their insightful comments and suggestions about our manuscript. In what follows we will present a point-by-point response to these questions and concerns.

  1. Most genes interact with other genes are from the same chromosome in cancers. Is this a cancer-universal phenomenon or breast cancer specific?

Yes, we have been analyzing for some time the observation (first documented around six years ago) regarding gene interactions within cancers, which are predominantly among genes on the same chromosome. We have been consistently and comprehensively researching possible mechanisms. This question sparked our analysis. Previous research from our group (Zamora-Fuentes, 2021, 2022; García-Cortés, 2023) consistently demonstrated the widespread loss of inter-chromosomal co-expression as a common trait across various cancer types. However, considering the distinct behaviors inherent to molecular subtypes in breast cancer, we opted to conduct a specialized analysis of the TF-target network within individual molecular subtypes.

  1. Why genes on the same chromosome often co-express and why loss of long-range co-expression patterns in cancers remain unclear?

The reason behind the frequent co-expression of genes located on the same chromosome and the persistence of long-range co-expression loss in cancers remains unclear. This is indeed the motivation behind this work. Although the exact cause remains elusive, its consistent occurrence across all examined cancer tissues suggests a potential physical or mechanistic nature, fundamentally reshaping the entire co-expression landscape. This scenario likely exerts influence on the expression patterns of numerous genes within cancer cells.

  1. Are the genes controlled by health TFs also controlled by the same set of TFs in BC? What kind of pathways are they involved in?

The answer, as far as we have observed, is no. The targets controlled by the TFs in the  healthy TF-target network are not shared with those regulated by TFs in cancer. In fact, the targets are also different between the subtypes. Derived from this result we decided to include a Venn diagram with the shared targets for each phenotype, as well as a brief description of the results. 

As the Reviewer may notice, the small amount of shared genes does not allow to perform an (statistically sound) enrichment analysis of them. However, it is interesting that similar TFs share a small set of targets. This is another instance of how the co-expression network structure determines the global gene regulatory landscape. Thank you for pointing this out.

Reviewer 2 Report

Comments and Suggestions for Authors

The study of Trujillo-Ortiz, et al. offered significant insights into the impact of transcription factors in the loss of inter-chromosomal co-expression for breast cancers. The authors developed gene regulatory networks for four distinct breast cancer types as well as healthy breast tissue. They thoroughly investigated the role of TF-target networks in influencing global co-expression patterns. Remarkably, the findings from this study suggest that transcription factors alone do not primarily contribute to the loss of inter-chromosomal co-expression. This discovery expands our comprehension of the regulatory mechanisms governing gene expression in breast cancer.

There are a few issues the authors should address:

1. The introduction is somewhat lengthy. Authors may want to consider condensing it to approximately half of its current length to enhance conciseness and clarity. For instance, mentioning TCGA as a source of expression data could be reserved for the Methods section.

2. Instead of using GENCODE v22 for the entire analysis, the authors initially annotated the TCGA samples with GENCODE v22 and subsequently filtered them based on v37. Further, it's worth noting that TCGA samples are currently annotated with v36 as an alternative approach.

3. Authors need provide the details in constructing TF-target network, including the source the TFs.

4. In line 241, the authors mentioned "A remarkable difference..." However, Figure 3A may not adequately illustrate this remarkable difference. It would be beneficial to include an additional figure to provide further support for this point. Additionally, consider enhancing the readability and comprehensibility of the y-axis for improved clarity.

5. Two typo in the manuscript, line 154 “to the tumors fro some of the same…” and line 357 “… RAD51 (Fig. 6)A. In addition, …”.

Author Response

The study of Trujillo-Ortiz, et al. offered significant insights into the impact of transcription factors in the loss of inter-chromosomal co-expression for breast cancers. The authors developed gene regulatory networks for four distinct breast cancer types as well as healthy breast tissue. They thoroughly investigated the role of TF-target networks in influencing global co-expression patterns. Remarkably, the findings from this study suggest that transcription factors alone do not primarily contribute to the loss of inter-chromosomal co-expression. This discovery expands our comprehension of the regulatory mechanisms governing gene expression in breast cancer.

 The authors want to thank Reviewer 2 for their dedicated review of our work. We have taken into account all of them when preparing a revised version of our work. In what follows we will address your review point-by-point.

There are a few issues the authors should address:

  1. The introduction is somewhat lengthy. Authors may want to consider condensing it to approximately half of its current length to enhance conciseness and clarity. For instance, mentioning TCGA as a source of expression data could be reserved for the Methods section.

 We have reduced the length of the introduction as requested. The revised version is much more concise, and the description of the TCGA database was moved to the methods section. Thank you.

  1. Instead of using GENCODE v22 for the entire analysis, the authors initially annotated the TCGA samples with GENCODE v22 and subsequently filtered them based on v37. Further, it's worth noting that TCGA samples are currently annotated with v36 as an alternative approach.

Thank you for noticing this. Perhaps we should have better clarified this in the manuscript. TCGA/GDC data was initially released with annotations via Gencode v22. This was our primary data source and decided not to re-annotate the full dataset –out of computational burden considerations–. Instead, we filtered out all protein coding genes annotated in Gencode v37 (that was the most up-to-date build at the time we did it) by retrieving the corresponding (harmonized) IDs in Gencode v22. This way we were able to recover an updated (i.e. v37) list of protein coding genes out of the original dataset (v22), without the need to re-annotate the enormous number of non-protein coding transcripts that we will not be using in our analysis anyways.

  1. Authors need provide the details in constructing TF-target network, including the source the TFs.

We have modified the Methods section, by including a brief section of the TF-target network construction, and also emphasizing the data source for transcription factors information. In the “Transcription factor binding motif analysis” subsection, we have provided a description of the data source for the list of TFs. 

  1. In line 241, the authors mentioned "A remarkable difference..." However, Figure 3A may not adequately illustrate this remarkable difference. It would be beneficial to include an additional figure to provide further support for this point. Additionally, consider enhancing the readability and comprehensibility of the y-axis for improved clarity.

We have modified that section of the results. We have provided a clearer description of the Y axis of Figure 3A in the figure caption. Additionally, we clarified the text of line 241. 

  1. Two typo in the manuscript, line 154 “to the tumors fro some of the same…” and line 357 “… RAD51 (Fig. 6)A. In addition, …”.

Done. Thank you for pointing this out.